# Formation of FePt–MgO Nanocomposite Films at Reduced Temperature

**Da-Hua Wei** [1,*](ORCID)**, Sheng-Chiang Chen** [1,†]**, Cheng-Jie Yang** [1,†]**, Rong-Tan Huang** [2,*]**, Chung-Li Dong** [3,*] **and Yeong-Der Yao** [4,*]

1    Department of Mechanical Engineering, Institute of Manufacturing Technology, National Taipei University of Technology (TAIPEI TECH), Taipei 10608, Taiwan; synder@ms6.hinet.net (S.-C.C.); bruce.60705@gmail.com (C.-J.Y.)
2    Department of Optoelectronics and Materials Technology, National Taiwan Ocean University, Keelung 20224, Taiwan
3    Department of Physics, Tamkang University, Tamsui 25137, Taiwan
4    Institute of Physics, Academia Sinica, Taipei 11529, Taiwan
*    Correspondence: dhwei@ntut.edu.tw (D.-H.W.); rthuang@mail.ntou.edu.tw (R.-T.H.); cldong@mail.tku.edu.tw (C.-L.D.); ydyao@phys.sinica.edu.tw (Y.-D.Y.)
†    These authors contributed equally to this work.

**Abstract:** The MgO nanolayer effect on the microstructure and magnetic characterizations added into Fe/Pt stacked films directly deposited onto MgO (001) single-crystal substrates at the reduced temperature of 380 °C using electron-beam technology was investigated in this present work. The nanograin isolation and exchange decoupling for the FePt–MgO system is attributed to the magnetic FePt isolated grains that originate from MgO atoms with a spreading behavior mostly along grain boundaries owing to its weaker surface energy than that of a single Fe or Pt element. The grain and domain size decreased when the MgO nanolayer was applied due to the interpenetration of MgO and created a strain-energy variation at the MgO/FePt interface. Measuring angular-dependent coercivity indicates a general trend of a domain-wall motion, and changes to the rotation of the reverse-domain model occurred as the MgO nanolayers were added into FePt films. The intergrain interaction is confirmed by the Kelly–Henkel plot, which shows that there is strong intergrain exchange coupling (positive δM type) between neighboring grains in the continuous Fe/Pt stacked films without MgO nanolayers. In addition, a negative δM type occurred when the Fe/Pt stacked films were added into MgO nanolayers, showing that the MgO nanolayer can be applied to adjust the force of intergrain exchange coupling between the adjacent FePt nanograins, and the addition of MgO nanolayers change into magnetic decoupling; thus, there was a formed dipole interaction in our claimed FePt–MgO composite structure of stacked ultrathin films at a reduced temperature of 380 °C.

**Keywords:** FePt nanocomposite; MgO nanolayer; stacked ultrathin films; magnetic decoupling/ isolation; angular-dependent coercivity; Kelly–Henkel plot

## 1. Introduction

Ordered FePt (CuAu (I)- $L1_0$ type) compound has undergone significant development over the past few decades due to its suitable material properties for magnetic storage media containing mostly grand-saturation magnetization ($M_s$~1100 emu/cc), a grand-anisotropy field ($H_i$~120 kOe), grand-energy products ($BH)_{max}$, a huge magnetocrystalline anisotropy constant ($K_u$~$10^8$ erg/cm$^3$), high Curie temperature ($T_c$~480 °C), and excellent environmental stability [1–13]. Mostly, the properties of high $K_u$ Fe-based compounds could retard the phenomenon of superparamagnetic effect and keep enough thermal stability to resist thermal fluctuation, even with a desirable grain size down to the nanometer scale, which means Fe-based alloys have future potential applications, such as in dense-ensemble spin systems such as electronic and magnetic storage devices with recording densities

beyond 10 Tbits/in$^2$ [14,15]. Production of the $L1_0$ ordered FePt composite films requires enough high-temperature processing (usually higher than 500 °C), such as post-deposition annealing or in situ substrate heating during disordered films deposition, to endure the activation energy barrier from a disordered face-centered cubic (fcc) to an ordered face-centered tetragonal (fct) $L1_0$ phase-transfer process. The ferromagnetic compound with a chemically ordered state usually displays a strong intergrain exchange coupling between the neighboring grains owing to the large-grain growth during the ordering/thermal process. Therefore, the nanocomposited and nanogranular structures formed at low-temperature processes have received significant attention due to the decoupling of the intergrain exchange coupling that could improve the signal-to-noise ratio; hence, they have been considered more suitable for next generation ultra-high-density magnetic storage media and novel miniature devices [16–27].

Reviewing previously published articles has shown the claims of the method of bottom or top additive layers; the doping effect of pure metal, nitride, or oxides is an effective way to adjust the crystalline orientation, microstructure, magnetic coupling, and chemical ordering of the $L1_0$ ferromagnetic thin films to satisfy the needs of the manufacturing process, especially in technologically ferromagnetic composites for modern multifunctional devices [28–48]. Insoluble oxide addition into magnetic FePt can provide a better grain/domain size control and with a weakened intergrain exchange coupling. In addition, MgO doped into the FePt alloys has been reported not only to enhance the ordering temperature, but also to cause the preferred crystal orientation with the magnetic easy axis to change from the out-of-plane to in-plane film direction [49–55]. For the purpose of decreasing the media noise, it is necessary to weaken the intergrain exchange coupling between the grains and suppress the grain growth in magnetic continuous media.

In this present work, a straightforward and simple method was demonstrated to obtain isolated FePt nanograins with (001) orientation when the addition of MgO nanolayers was used to form FePt self-organized nanogranular films for ultra-high-density magnetic storage media. This study also shows significant differences without and with MgO additive nanolayers into FePt stacked ultrathin films on the microstructure and magnetic characteristics at a reduced deposition temperature of 380 °C. Compared with our previous works, without any buffer layer, a thinner thickness of an Fe and Pt stacked layer were designed and assisted in this present study to decrease the formation temperature of the $L1_0$ ordered FePt composite transformation [56,57]. The corresponding magnetization reversal mechanism and intergranular exchange coupling of the claimed FePt–MgO composite structure with stacked ultrathin films were also systematically investigated. Particularly, the relationship between micro/nanostructure and the magnetization reversal process (coercivity mechanism) is discussed in detail for $L1_0$ ordered FePt–MgO films and this similar concept could become the viewing reference point for recent developments in the general design rules and specific technical methods in related anisotropic alloy nanocomposite rare earth permanent films.

## 2. Experiments and Composite Film Structures

Nanocomposited Fe/Pt stacked structures made up of [Fe (0.5 nm)/Pt (0.5 nm)]$_{10}$ ultrathin films were deposited under a vacuum of $6.67 \times 10^{-6}$ Pa by electron-beam technology (homemade) directly on the (001) magnesium oxide substrates without any buffer layer. Two MgO layers were symmetrically evaporated atop the [Fe/Pt]$_3$ and [Fe/Pt]$_7$ bilayers without introducing any oxygen gas, and their thickness was fixed at 1 nm. All films were fabricated at a reduced temperature of 380 °C with an evaporation rate of around 0.02 nm/s. The preparation process of Fe/Pt stacked ultrathin films in this study was with the purpose of lowering the diffusion distance of Pt and Fe atoms into the $L1_0$ ordered crystal lattice, whose concept is the same as the atomic arrangement in the unit cell via an artificial atomic-scale stacked formation [58–60]. The field emission electron probe X-ray microanalysis (FE-EPMA, JEOL, Tokyo, Japan) was used to confirm the chemical composition of the binary FePt alloy, which was Fe$_{48}$Pt$_{52}$. The X-ray diffraction (XRD, PANalytical,

Almelo, The Netherlands) with Cu $K_\alpha$ radiation (λ = 1.54 Å) was used to identify the crystalline structure, and the receiving slit was set to 0.1 mm, and the time per step was 3 s with a scan speed of 0.01° $2\theta$/s during the XRD measurement with the proportional counter. The Zeiss Supra field emission-scanning electron microscope (FE-SEM, Dresden, Germany), equipped with an Oxford Instruments NordlysNano[TM] camera was used to observe the surface microstructure of the films. It could provide a sufficient number of grains for the initial estimate of grain size in the scan area with $3 \times 3$ μm$^2$, and the beam was with a step-moved size of 10 nm. A plane-view microstructure was used to measure the grain orientation distribution and grain size distribution quantitatively. The distinguishing feature of transmission electron microscopy (TEM, JEOL, Tokyo, Japan) was used to observe the crystalline nanostructure and corresponding nanograin sizes, respectively. The vibrating sample magnetometer (VSM, Lake Shore 7400, Westerville, OH, USA) with a maximum applied field of 20 kOe was used to identify the magnetic characteristics at room temperature. The Dimension 3100 atomic force microscope (AFM) equipped with magnetic force microscopy (MFM) mode (Veeco Instruments) has been used to observe the surface roughness and magnetic domain morphology.

The following comparison will be focused on the pure Fe/Pt stacked film structures without and with addition of two period MgO nanolayers (hereafter denoted as FePt–MgO) in order to investigate the variable defects at the interfaces between MgO and Fe/Pt stacked ultrathin films on the microstructure and corresponding magnetization reversal mechanism of the claimed FePt–MgO composited nanogranular films.

## 3. Results and Discussion

Figure 1 shows the FE-SEM surface micrograph of the FePt films (a) without and (b) with a total of 2 nm additive MgO layer obtained by secondary electron image (SEI) mode, respectively. Pure FePt film without the MgO additive nanolayer was connected, and the microstructure seems like a continuous film, as shown in Figure 1a. In addition, the surface micrograph changes from continuous to nanogranular microstructures in the FePt with the addition of MgO nanolayers as shown in Figure 1b. It confirms that the continuous formed state of the pure FePt film is disrupted by the addition of the MgO nanolayers, leading to the generation of nanogranular FePt composited thin films.

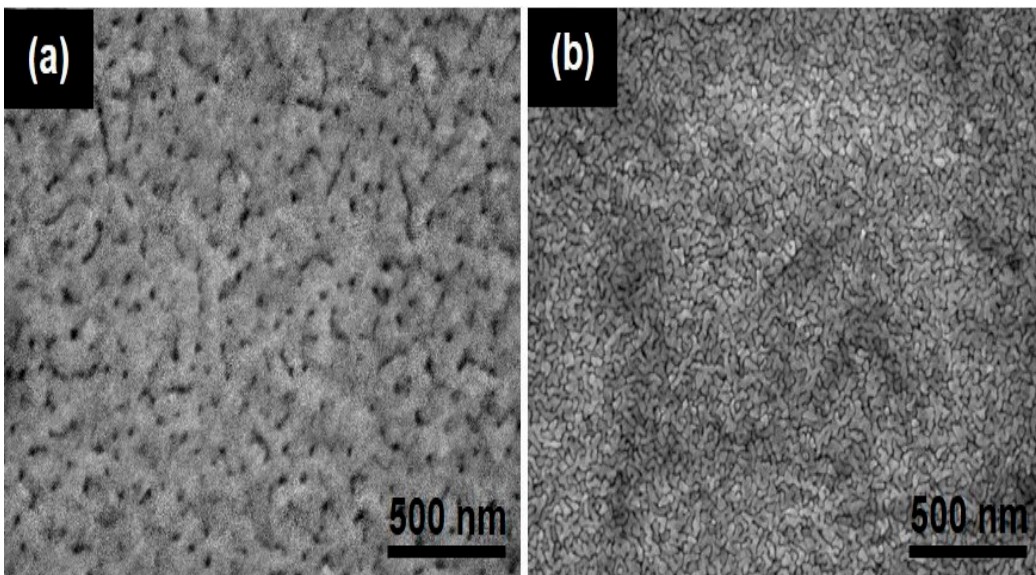

**Figure 1.** FE-SEM surface micrographs of the FePt films (**a**) without and (**b**) with a total of 2 nm MgO additive nanolayers were obtained by secondary electron image (SEI) mode, respectively.

Figure 2 shows the bright-field TEM images of the FePt films (a) without and (b) with MgO additive nanolayers. TEM analysis was employed to examine the crystallographic

orientation and microstructure of the films in detail. A clear variation in the surface micrograph of the FePt is displayed in the structures with MgO additive nanolayers. The surface micrograph obviously varies from continuous to nanogranular structures as the MgO nanolayers are added into the FePt. The images reveal that the average grain size of FePt and FePt–MgO are about 15 and 9 nm, respectively. It can be observed that the reduced grain size tends to be more uniform with addition of the MgO. The FePt films were grown initially with the island growth mode at this total film thickness [61]. The strongly faceted islands of FePt grains are observed with large size distribution. This implies that a number of small grains were formed in the initial stage of the film deposition, and then they connected to form big grains with continuous morphology. The primary facet planes of FePt are (100) and (010) and alignment with the direction of MgO [100], and the minor facet plane is (110), indicating that the surface energy of the (100) planes is the lowest. The stripe contrast observed in the FePt–MgO grains are the Moiré pattern generating from the lattice parameter difference between FePt and MgO, which is generated from the lattice mismatch between FePt and MgO with $a_{FePt} = 0.40$ nm and $a_{MgO} = 0.42$ nm, respectively [62]. Figure 2b shows that the FePt grains are dispersed in the MgO matrix. The reduction of the grain size should be mainly due to the MgO additive nanolayers/grains between the Fe and Pt layers/grains acting as barriers, which have prevented the diffusion and migration of the Fe and Pt atoms, and thus the coarse grain is limited. The surface energy of Fe (2.9 $Jm^{-2}$) and Pt (2.7 $Jm^{-2}$) are much greater than that of MgO (1.2 $Jm^{-2}$) [63]. This describes that MgO atoms could lightly diffuse into the magnetic FePt grains along their boundaries and create a strain-energy variation at the interface due to its much smaller surface energy than that of pure Pt or Fe atoms [64].

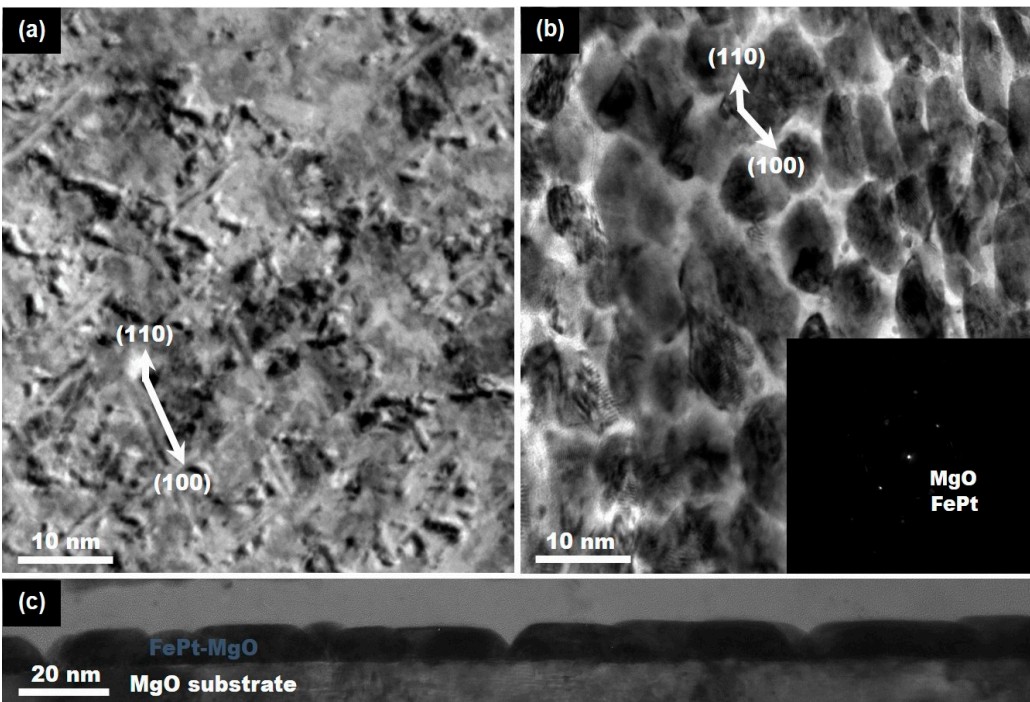

**Figure 2.** Representative TEM bright-field micrographs for the FePt films (**a**) without and (**b**) with MgO additive nanolayers, and the inset is the corresponding selective electron diffraction pattern (SEDP), respectively. (**c**) is the cross-sectional TEM micrograph for the FePt–MgO composited thin films.

The inset shown in Figure 2b is the electron diffraction pattern (EDP) of FePt–MgO films. The selected area EDP shows that the orientation relationship between the FePt grains and the MgO substrate is along the MgO [100]//FePt [100] direction, indicating that the FePt–MgO films are (001) preferred orientation even after the FePt with the addition of

MgO process. In both FePt and FePt–MgO systems, highly textured (001) FePt films with a $L1_0$ crystal structure were formed. In the case of the FePt–MgO system, the presence of twins will introduce some local misalignments of the growth axis from the [001] direction. The surface roughness will be greater in the FePt–MgO system due to existing MgO/FePt interfaces and the thicker FePt–MgO composited film, indicating a more severe intermixing of the FePt–MgO alloy composite. Figure 2c shows cross-sectional TEM micrographs of nominally 12 nm thick FePt–MgO films grown on MgO(001) substrate. Remarkably different growth morphology is obtained and confirmed in the FePt–MgO system. Typically, the pure FePt with 10 nm thick film is continuous. In the case of the FePt–MgO film, the film micrograph has partially coalesced elongated islands with average real thickness equal to $11 \pm 1$ nm. This behavior shows an island growth case whereby the initially grown grains extend laterally with increasing thickness, and finally coalesce into a continuous state. Meanwhile, MgO nanolayers added into FePt that formed inclined twins and stacking faults on the {1 1 1} plane, thus causing the disrupted grain growth which is presented here.

The out-of-plane (perpendicular) and in-plane (parallel) hysteresis curves for FePt films without and with MgO additive nanolayers are shown in Figure 3a,b, respectively. The magnetic easy axis is constantly perpendicular to the film plane, and the perpendicular anisotropy is clearly demonstrated for both films. The related magnetic characteristics including remanent squareness ratio ($M_{r\perp}/M_{s\perp}$), saturation magnetization ($M_{s\perp}$), and out-of-plane coercivity ($H_{c\perp}$) values are listed in Table 1 in detail. The coercivity value of the FePt thin films reduced from 7500 to 5100 Oe (FePt–MgO). The saturation magnetization ($M_{s\perp}$) and remanent squareness ratio ($M_{r\perp}/M_{s\perp}$) values were reduced and ranged from 825 to 687 emu/cm$^3$ (FePt–MgO) and 0.99 to 0.91 (FePt–MgO), respectively. The reduction of the squareness ratio may indicate that the intergranular interactions of FePt are less magnetically coupled with the addition of the MgO nanolayer and could prove that MgO atoms penetrated into FePt magnetic grains through the grain boundary to decouple the intergranular interaction between the FePt neighboring magnetic grains. The magnetic characteristics including the magnetic reversal mechanism and the corresponding intergranular exchange coupling of the designed FePt–MgO composite-stacked ultrathin films are compared and discussed below. The difference of magnetic characteristics for FePt–MgO may be due to the addition of MgO acting as nucleation sites to prevent the domain-wall motion and varying the coercivity. The other difference may be induced by the interlayer diffusion that changes the film composition, thus decreasing the film coercivity of FePt–MgO. Actually, the coercivity of $L1_0$ FePt alloy is strongly dependent on the degree of chemical ordering. Comparison with the XRD results will be discussed below, and the variation of FePt film coercivity with the MgO additive nanolayers is consistent with the variation of the chemically ordering degree.

The angular dependence of coercivity is for the purpose of examining the magnetization reversal behavior of the FePt films without and with MgO additive nanolayers, as shown in Figure 4. Shown in Figure 4 are the ideally theoretical curves, defining two boundary conditions of rotation of the Stoner–Wohlfarth (S–W) and domain-wall motion types, respectively. For an ideal domain-wall motion type, the coercivity at the angle $\theta$ is proportional to $1/\cos(\theta)$, where $\theta$ is the angle between the easy axis of the uniaxial magnetic anisotropy and its applied field. For the S–W type with the rotation mechanism, the transformation of the coercivity reduces with enhancing $\theta$. The angular dependence of the coercivity curve for the FePt without the MgO nanolayer showed typical peak behavior owing to the continuous film micrograph that existed in the pure FePt films. In this present work, the magnetic alignment of the easy axis perpendicular to the film plane is near to the Bloch-like domain walls. This significantly enhances the propagation of the domain walls while the surface micrograph of pure FePt films is continuous, as shown in Figures 1 and 2. When the FePt films were added with MgO nanolayers, the curve was near to the rotation type, and the magnetization reversal behavior became more independent. The above results indicate an inclination in progress lessened domain-wall motion behavior, but an enhanced rotation type in the magnetization reversal behavior with the addition of the MgO layer into the FePt films, which may reduce the

intergranular interaction between the FePt neighboring magnetic grains. As for the obtained results mentioned above, the magnetization reversal mechanism and corresponding coercivity mechanism of the FePt–MgO composite alloy case could be simply adjusted by the total content of the MgO nanolayer.

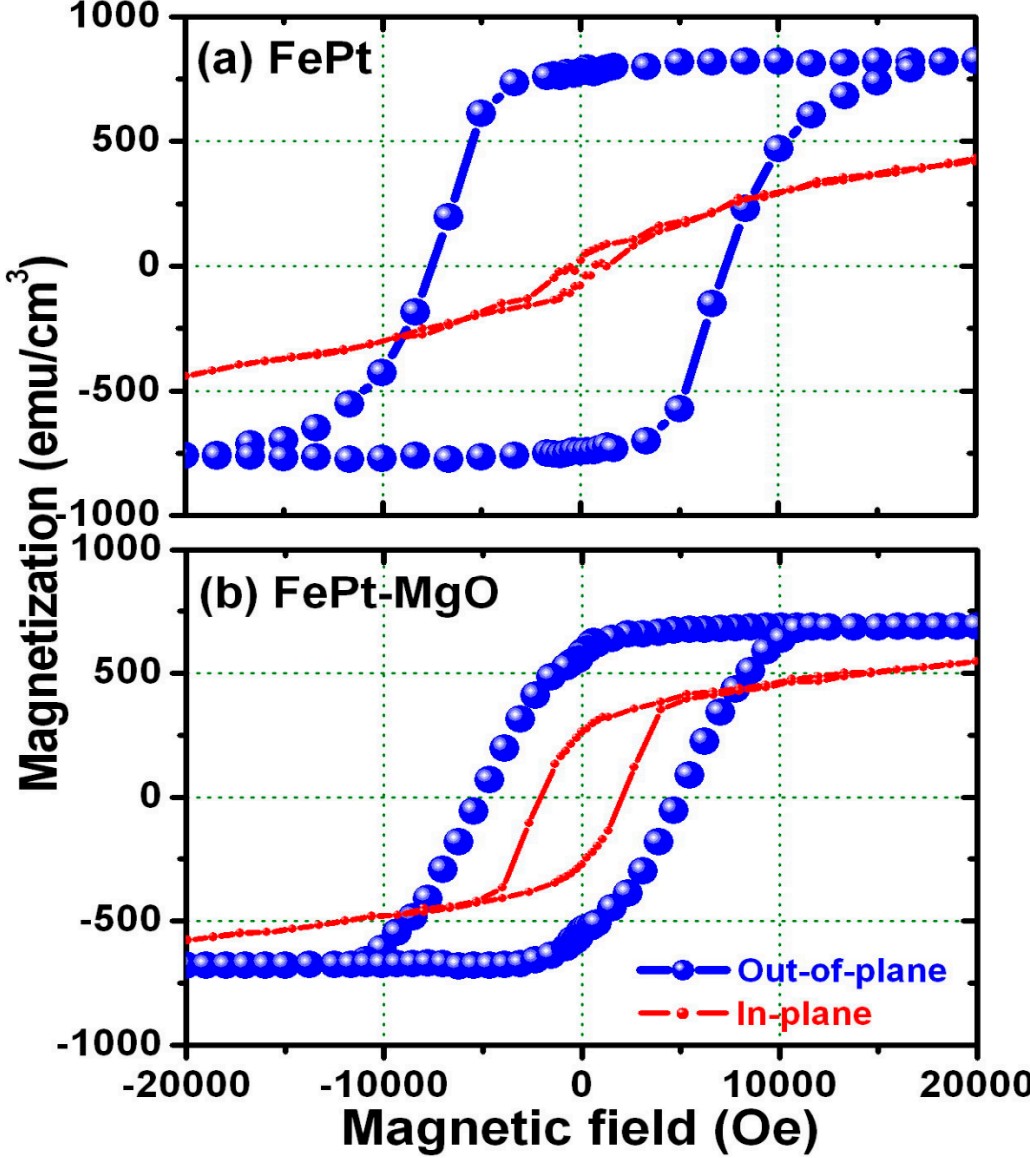

**Figure 3.** The out-of-plane (blue) and in-plane (red) hysteresis curves for the FePt films (**a**) without and (**b**) with MgO additive nanolayers, respectively.

**Table 1.** FePt films without and with MgO additive layers listed the out-of-plane coercivity ($H_{c\perp}$), saturation magnetization ($M_{s\perp}$), and remanent squareness ratio ($M_{\mathrm{r}\perp}/M_{s\perp}$) values, respectively.

| MgO Thickness (nm) | $H_{c\perp}$ (Oe) | $M_{s\perp}$ (emu/cm$^3$) | $M_{\mathrm{r}\perp}/M_{s\perp}$ (Ratio) |
|---|---|---|---|
| 0 | 7500 | 825 | 0.99 |
| 2 | 5100 | 687 | 0.91 |

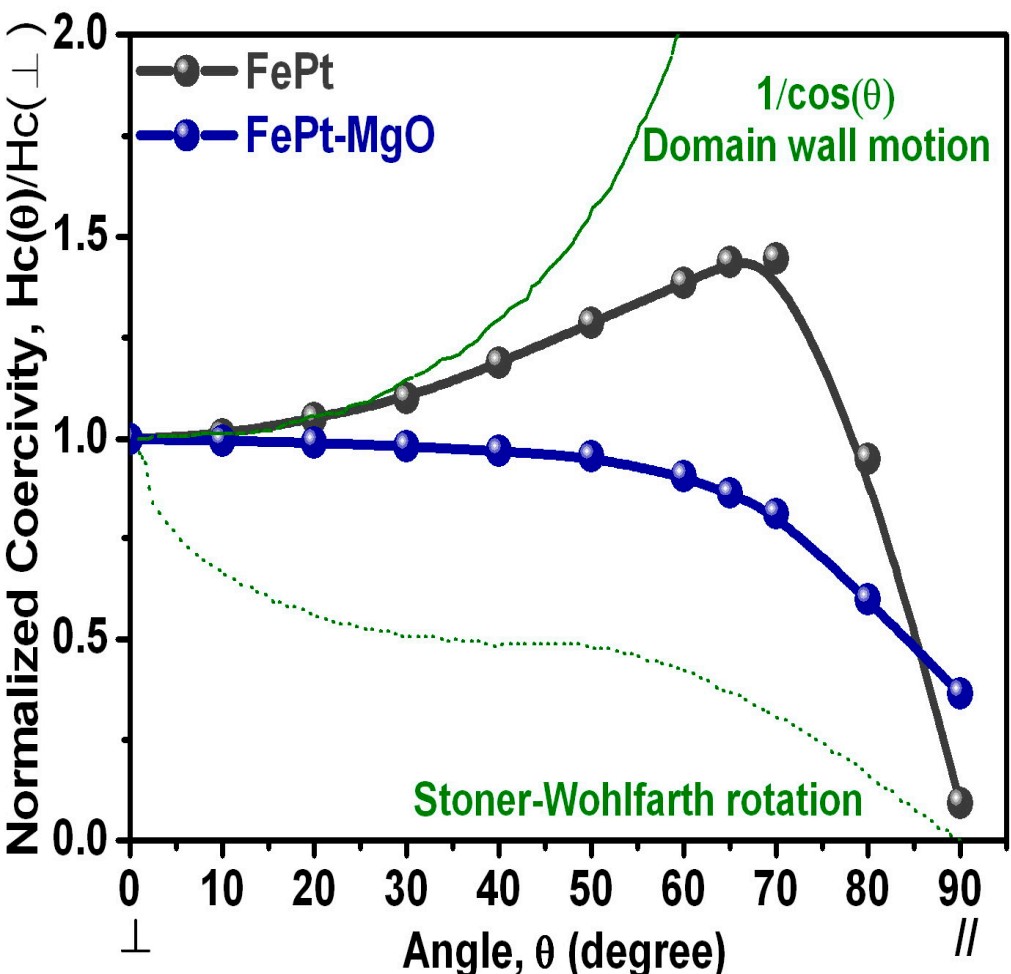

**Figure 4.** The curves for angular dependence of coercivity for the FePt films without and with MgO additive nanolayers, respectively. The angle refers to that between the easy axis (film normal) and the applied magnetic field direction.

Figure 5 shows a Kelly–Henkel plot ($\delta M$ measurement) for the FePt films without and with MgO additive nanolayers, respectively. The $\delta M$ measurement has been applied to distinguish the intergrain interaction in magnetic materials, which is defined as [65]:

$$\delta M = M_{DCD}(H) - [1 - 2M_{IRM}(H)] \tag{1}$$

where $M_{DCD}(H)$ and $M_{IRM}(H)$ are the normalized dc-demagnetization remanence and isothermal remanence as a function of the applied magnetic field, respectively. The positive $\delta M$ type shows strong ferromagnetic intergrain interactions. In addition, the negative $\delta M$ type shows dipole intergrain interactions associated with incoherent rotation. It can be clearly observed from Figure 5 that FePt films without MgO addition showed a positive $\delta M$ type (strong ferromagnetic intergrain interaction), while FePt films with the MgO additive nanolayers showed only the negative $\delta M$ type at all applied magnetic fields (decouple interaction). This describes that the independent moment rotation of the FePt films is caused by the MgO atoms being penetrated into FePt magnetic grains via the grain boundary, leading to the reduction of exchange intergrain interactions between neighboring magnetic grains. The important parameter $\delta M$ value is well known to reflect the noise of magnetic recording media; this value variation can be adjusted easily by the total content of MgO in our claimed FePt–MgO composite case, which decides the intergrain interaction for the ferromagnetic composite alloy system.

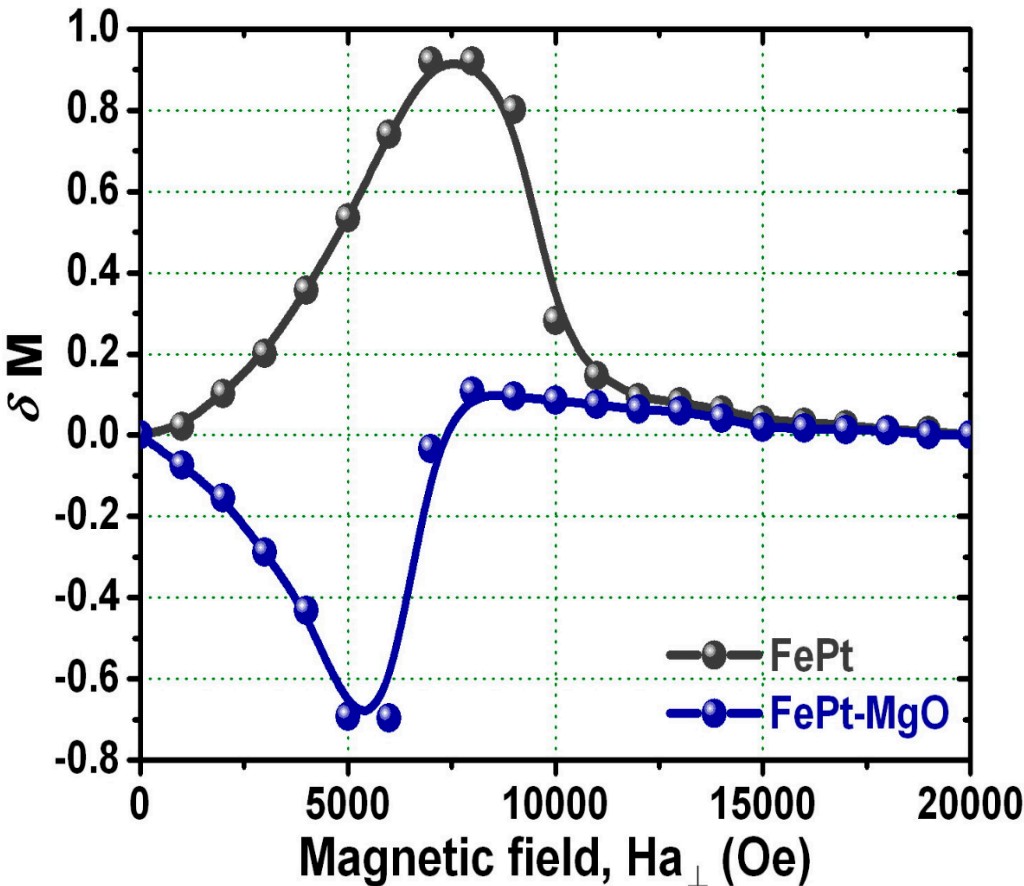

**Figure 5.** *δM* measurement of the FePt films without and with MgO additive nanolayers are with the external field applied perpendicular to the film plane direction.

The obvious change of the perpendicular magnetization behavior in FePt films without and with MgO additive nanolayers is clearly identified by the initial magnetization loops as shown in Figure 6a. The normalized initial magnetization loop will be used to clarify the magnetization reversal mechanism as shown in Figure 6b. For the FePt films with the addition of MgO nanolayers, it became much harder to saturate compared with the pure FePt films at the same applied magnetic field. If the magnetization reversal behavior is near to rotation of the Stoner–Wohlfarth (S–W) type, the single domain magnetic grains only reversed their magnetization behavior when the external applied magnetic field exceeded the anisotropy energy [66–70]. Thus, the pure FePt film without addition of MgO nanolayers is close to the domain-wall motion type of initial magnetization loops, and FePt film with addition of MgO nanolayers is close to the typical nucleation type of the initial magnetization loop.

The coercivity of ferromagnetic substances is strongly correlated with their anisotropy field and microstructure. Consequently, the temperature dependence of coercivity, $H_c(T)$ of a ferromagnetic material can be described as [71]:

$$H_c(T) = \alpha_K \alpha_{ex} H_i(T) - N_{eff} M_s(T) \qquad (2)$$

where $\alpha_K$, $\alpha_{ex}$, and $N_{eff}$ are represented by microstructure parameters and are associated with the nonideal microstructure of the ferromagnetic material. $H_i$ is the anisotropy field and $M_s$ is the saturation magnetization. The parameter $\alpha_K$ indicates the effect of the nonideal surface micrograph of grains on the crystal anisotropy. The parameter $\alpha_{ex}$ considers the effect of the exchange coupling between neighboring FePt nanograins related to the δM measured type. The effective demagnetization factor $N_{eff}$ is caused by enhanced stray fields at the corners and edges of the magnetic grains. We believe that the microstructure

($\alpha_{ex}$) and anisotropy field ($H_i$) parameters should act as the effective roles related to the coercivity variation with the addition of MgO into FePt in this composite case. Our claimed FePt–MgO (001) nanogranular film with large coercivity (5.1 kOe) satisfies the requirements for the future application of high-density magnetic storage devices.

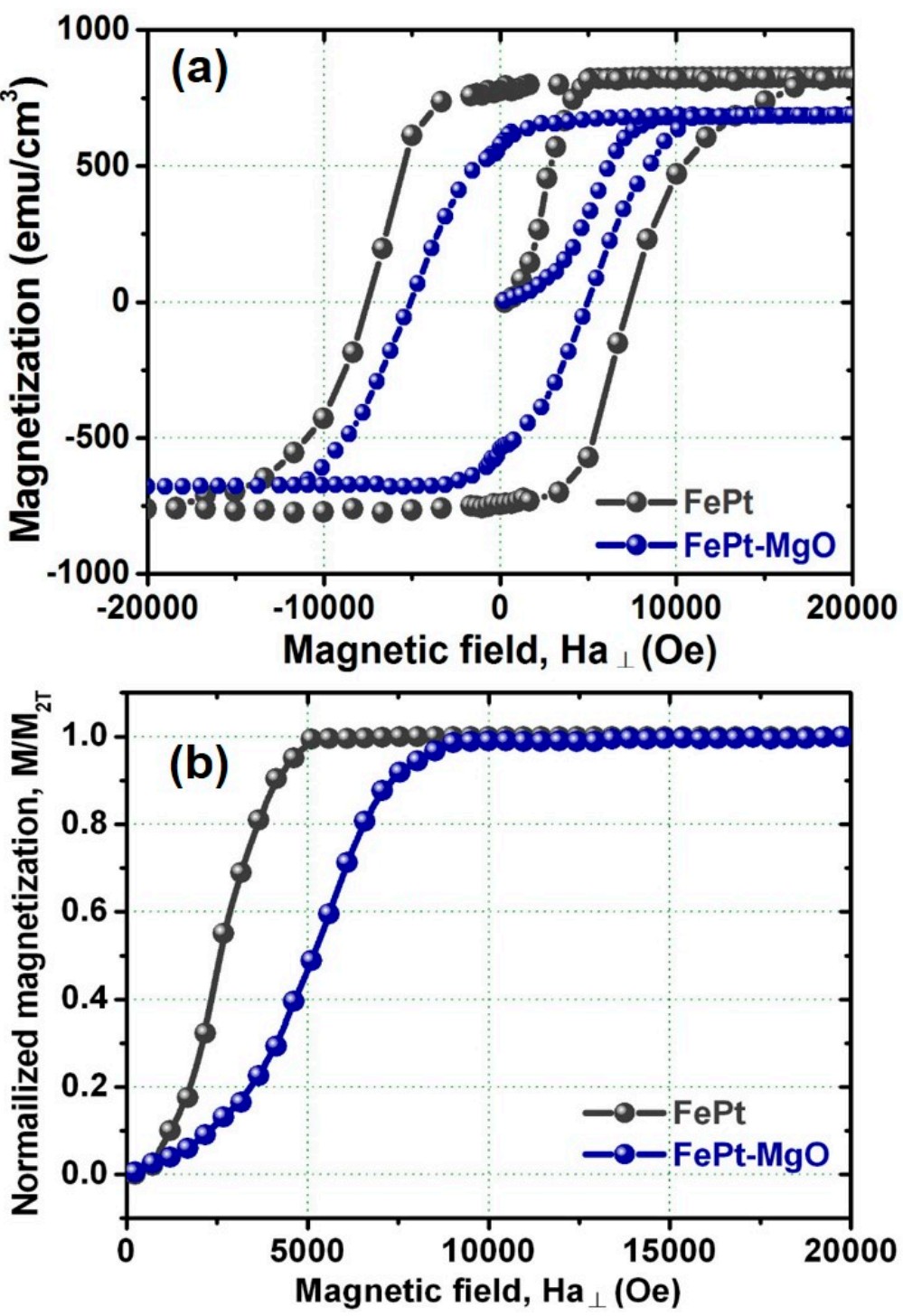

**Figure 6.** (**a**) Perpendicular magnetization loops and corresponding (**b**) normalized initial magnetization loops for the FePt films without and with MgO additive nanolayers. The magnetic field was externally applied in the perpendicular direction to the FePt and FePt–MgO both cases.

Figure 7 shows the surface micrograph and magnetic domain structures of the FePt films (a,c) without and (b,d) with MgO additive nanolayers, which were directly deposited

on MgO substrates and measured with the AFM and MFM modes. The surface micrograph of the pure FePt film shows the continuous film micrograph, which is consistent with FE-SEM and TEM images as shown in Figures 1 and 2. It indicates that the interconnected films formed by connected-together grains, and generated adjacent magnetic domains in the pure FePt films as shown in Figure 7a. While adding the MgO nanolayers, the grain micrograph in the AFM image changes from the continuous film interconnection to a nanogranular structure, which is much smaller than that observed in pure FePt film as shown in Figure 7b. The addition of the MgO nanolayers should have prevented the interconnection FePt network from being formed. It indicates that the continuous lateral growth of the FePt film is disrupted by the addition of the MgO nanolayers and leads to the generation of the nanogranular structure. The ac-demagnetized MFM signal image was performed from the same area that was measured by the AFM equipment. Compared to the domains obtained in the MFM images as shown in Figure 7c,d, the magnetic domain size is much greater than the grain interconnection obtained in the AFM image. In addition, the mazelike domain was observed in pure FePt film, and the domain size was reduced with the enhancing of the MgO content.

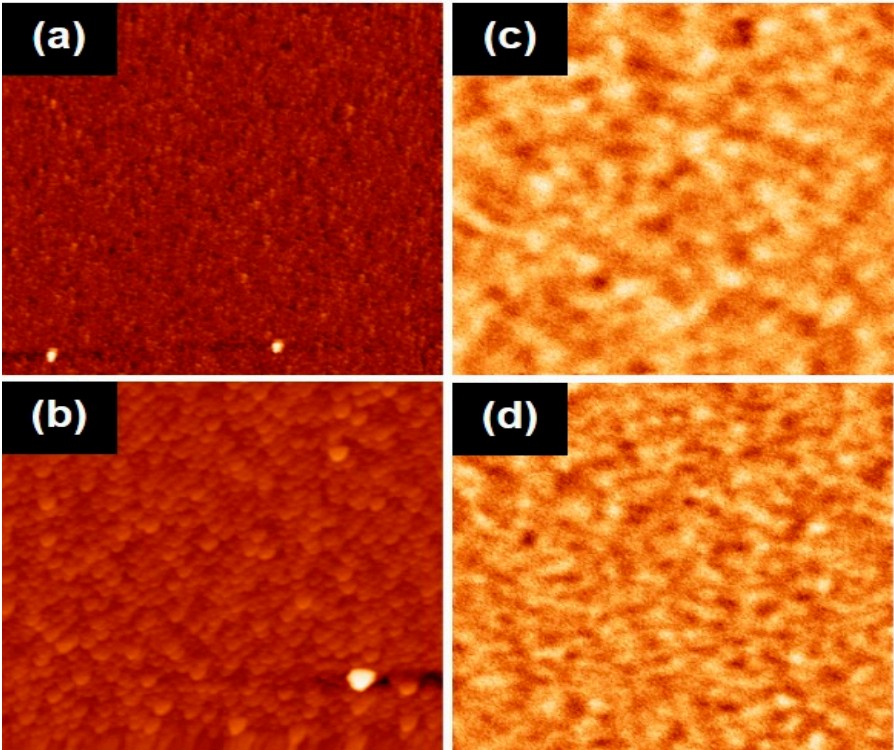

**Figure 7.** Surface morphology of the FePt films (**a**,**c**) without and (**b**,**d**) with MgO additive nanolayers, respectively, which were measured by the AFM (**left**) and MFM (**right**) modes. The scan area is $2 \times 2$ μm$^2$.

At the same time, most of the boundary between the grains in the AFM still existed and were observed in the MFM image. This suggests that most of the grains in the FePt–MgO film are noninteraction single domains. The noninteraction nanogranular films are very close to the Stoner–Wohlfarth domain type [72]. Thus, the magnetization reversal mechanism could be controlled and adjusted from the domain-wall motion in pure FePt film to a nanogranular single domain rotation mechanism for our claimed FePt–MgO case as shown in Figure 4. The coercivity of the FePt–MgO composite case may be attributed to the non-interconnection mechanism of the single domains, although the coercivity value is still smaller than that of the theoretical calculation from the Stoner–Wohlfarth type. The magnetic hard FePt-based films with high magnetocrystalline anisotropy have been attracting much attention for application in modern magnetic devices since their critical

magnetic grain size for the superparamagnetic limit can be reduced to the nanometer scale. It is well known that a finer magnetic cluster diameter is favorable for the ultra-high-density magnetic storage media.

Figure 8 shows the in-plane XRD patterns for FePt film structures (a) without and (b) with MgO additive nanolayers, respectively. Figure 8c,d is the related slow $\theta$–$2\theta$ scan curves of the FePt(002) peak in Figure 8a,b, respectively. In addition to the mainly (002) peak, $L1_0$-ordered (001) superlattice peaks of the FePt phase were clearly observed for both films. The unlabeled sharp peaks are owing to the MgO substrate. Only (00$n$) diffraction peaks in Figure 8 were observed in the whole diffraction patterns ($\theta$–$2\theta$ scan) with a wide scanning range, evidencing that all FePt films without and with MgO additive nanolayers have been strongly textured to the (001) planes, and also supporting that the stacked ultrathin film structures were epitaxially deposited on the MgO substrate. The intensities of the (002) fundamental and (001) superlattice peaks of the FePt were reduced for FePt films with MgO additive nanolayers, evidencing the ordering degree of FePt–MgO composite structured film is influenced by the MgO additive nanolayers. The FePt peaks are relatively wide for the FePt films with MgO additive nanolayers compared with the pure FePt. On the other hand, the full width at half maximum (FWHM) of the slow scan peaks of FePt (002) for FePt with the MgO additive nanolayers is greater than that of pure FePt without MgO, as shown in Figure 8c,d, showing that the lattice deformation of the FePt films is caused by inhomogeneous solidification of MgO owing to immiscibility of MgO in the FePt crystal lattice. These results imply that MgO atoms tend to diffuse into the FePt alloy via the grain boundary to slightly widen the peak curve of FePt (002). According to the obtained results mentioned above, which describe that the grain size of the FePt binary alloy is reduced with the addition of MgO into the FePt phase, this fact is consistent with the FE-SEM, TEM, and AFM observations.

The disorder–order transformation is dominantly varied by the growth process of $L1_0$-ordered grains that have been reported [73]. The activation energy not only for grain growth but also for disorder–order transformation plays the important role of the driving force in the FePt alloy. Hence, the grain growth will be suppressed by the $L1_0$-ordering process. In addition, it has been reported that the ordering process of Fe-based thin films could be adjusted by atom diffusion [74–77]. The above results indicate that MgO could partially penetrate into the FePt films, leading to the lattice deformation of FePt structures consistent with the widening curves of the FePt (002) peak. In this present work, the effects of additive MgO nanolayers on the magnetic behavior and corresponding magnetization reversal mechanism into FePt films were displayed and compared, without any fabrication condition in our claimed FePt–MgO composite structure being varied except for the pure FePt stacked ultrathin film structures without and with addition of MgO nanolayers.

X-ray reflectivity patterns for the FePt films without and with MgO additive nanolayers are shown in Figure 9a,b, respectively. It is clearly observed that the amplitude intensities of the oscillation fringes (Kiessig fringes) of FePt added with MgO nanolayers are getting lower than those without MgO nanolayers. This also implies that MgO mainly diffuses into the FePt films along the grain boundaries, and the microstructure refining effect could be obtained via our claimed additive method of MgO nanolayers into the FePt alloy films.

FePt composited films fabricated at room temperature are a disordered ($A1$) phase with a low cubic magnetocrystalline anisotropy. The formation of the ordered ($L1_0$) phase usually needs high-temperature treatment (beyond 500 °C). Our work claims a multilayer method with the addition of MgO nanolayers into FePt stacked ultrathin films is different from the method of co-sputtering or the co-evaporation technique, and better in obtaining the *c*-axis to be highly oriented and perpendicular to the film plane at the reduced temperature of 380 °C, which is suitable for future applications in high-density perpendicular magnetic recording media.

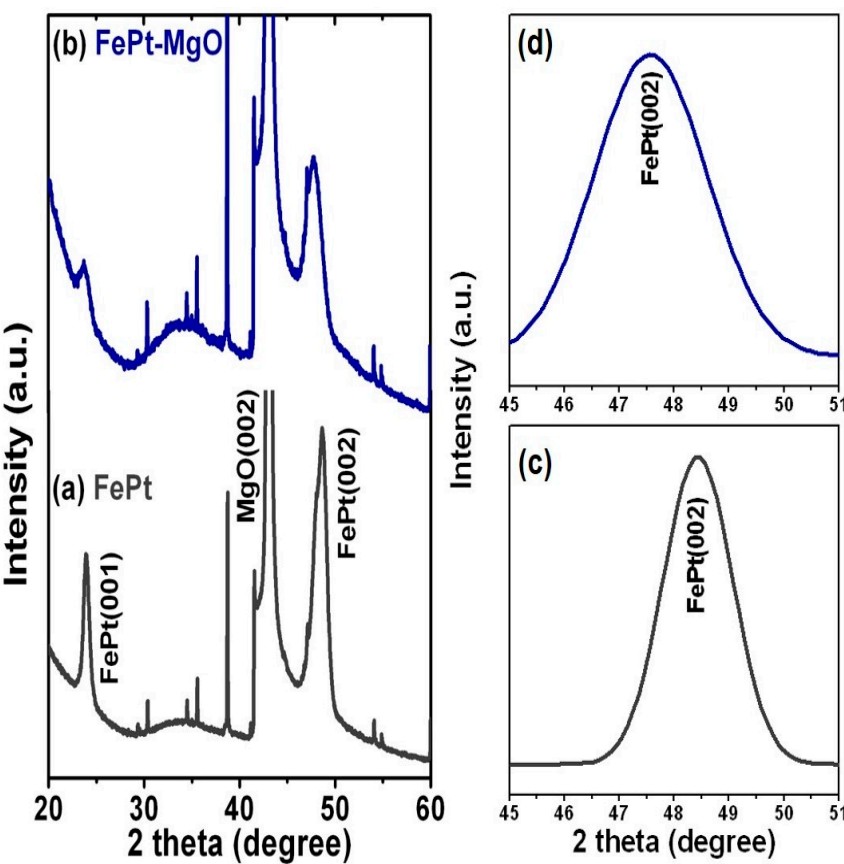

**Figure 8.** X-ray diffraction patterns ($\theta$–$2\theta$ scans) for the FePt films (**a**) without and (**b**) with MgO additive nanolayers, respectively. The related slow scan peak curves of the FePt (002) in the $\theta$–$2\theta$ scan (**c**) without and (**d**) with MgO additive nanolayers, respectively.

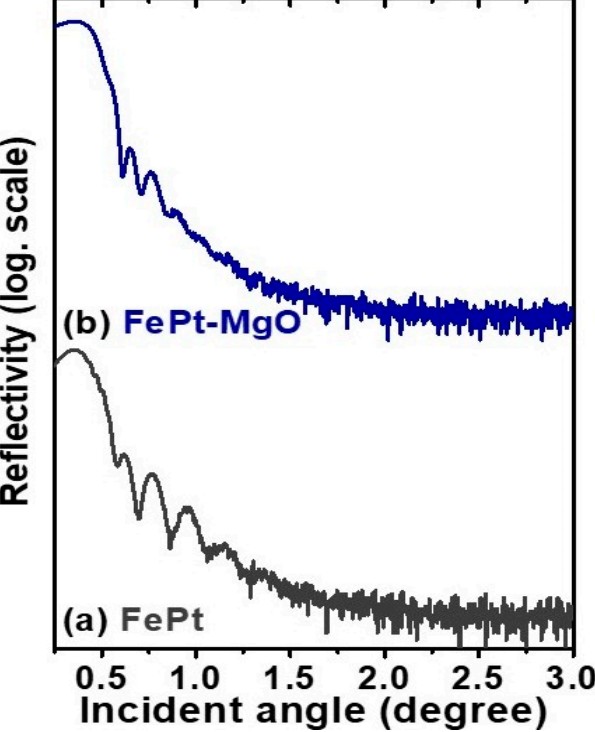

**Figure 9.** X-ray reflectivity patterns for the FePt films (**a**) without and (**b**) with MgO additive nanolayers, respectively.

## 4. Conclusions

In this article, a straightforward and simple method is claimed, which showed that addition of MgO nanolayers into FePt stacked ultrathin films could weaken the intergrain exchange coupling and, thus, provide enough coercivity (5.1 kOe) and satisfy the requirements for the modern media devices at the reduced deposition temperature of 380 °C. The angular dependence of the coercivity measurement indicated that with addition of MgO nanolayers into the FePt stacked ultrathin films, the magnetization reversal mechanism was observed to adjust from the domain-wall motion type to be closer to the rotation type that dominated in the FePt–MgO composite films. Thus, the FePt composited film with addition of MgO nanolayers is effective to act as magnetic nucleation sites in the FePt films, weaken the intergrain coupling strength, and reduce media noise, which will be a great advance to develop FePt-based heat-assisted magnetic recording (HAMR) media with enhanced signal-to-noise ratio in the development of modern applications of ultra-high-density perpendicular spin electronic nanodevices.

**Author Contributions:** Conceptualization, D.-H.W. and Y.-D.Y.; methodology, D.-H.W.; software, D.-H.W., S.-C.C. and C.-J.Y.; validation, D.-H.W. and Y.-D.Y.; formal analysis, D.-H.W. and Y.-D.Y.; investigation, D.-H.W. and Y.-D.Y.; resources, Y.-D.Y.; data curation, D.-H.W., S.-C.C. and C.-J.Y.; writing—original draft preparation, D.-H.W.; writing—review and editing, D.-H.W. and Y.-D.Y.; visualization, D.-H.W.; supervision, D.-H.W. and Y.-D.Y.; project administration, D.-H.W. and Y.-D.Y.; funding acquisition, D.-H.W., R.-T.H., C.-L.D. and Y.-D.Y. All authors have read and agreed to the published version of the manuscript.

**Funding:** This research was funded by the Ministry of Science and Technology (MOST), through grant numbers 111-2731-M-027-001 and 108-2628-E-027-002-MY3 and the University System of Taipei Joint Research Program through grants no. USTP-NTUT-NTOU-111-02.

**Acknowledgments:** The authors acknowledge financial support of the main research projects of the Ministry of Science and Technology (MOST) under grant nos. 111-2731-M-027-001 and 108-2628-E-027-002-MY3. Da-Hua Wei and Rong-Tan Hang greatly appreciate the financial support of the University System of Taipei Joint Research Program (grant no. USTP-NTUT-NTOU-111-02).

**Conflicts of Interest:** The authors declare no conflict of interest.

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
