# Peer review of "Formation of FePt–MgO Nanocomposite Films at Reduced Temperature"

_jcs, doi:10.3390/jcs6060158_

Round 1
Reviewer 1 Report
The peer-reviewed paper presents interesting results and develops an approach that provides the necessary amount of coercive force required for modern memory devices with recurrent recording.
The manuscript can be accepted for publication after some revision containing answers to a number of questions and comments.
- Abstract, introduction and following text mention ‘a high exchange-coupled interaction between the neighboring grains’ ‘strong exchange coupling (positive δM) between neighboring grains in the FePt continuous films’, ‘intergrain exchange coupling between the adjacent FePt nanograins’, and ‘decoupling of the intergranular interaction'. At the same time, the ‘existence of dipole interaction in our designed FePt-MgO composite structure' is discussed. It is necessary to explain more clearly the terminology used here, whether two types of interaction having a different nature are meant (dipole-dipole magnetostatic interaction vs more long-range exchange interaction between individual grains which can hardly be considered reliably established).
- In the introduction it was stated ” Particularly the relationship between micro-/nanostructure and the magnetization process (coercivity mechanism) is discussed in detail for L10 ordered FePt-MgO films and related rare earth–transition alloy composited films”, but nowhere else is it said about composite films containing rare earth elements. Apparently, alloys based on transition metals are assumed here, but not RE elements, but Pt.
- Has the hysteresis loops in Fig.3 been corrected to account for the demagnetizing field of the continuous film when the films are magnetized in the perpendicular direction?
- Is there a difference in the magnetization curves depending on the direction of the magnetic field in the plane of the film? This would help to identify a possible magnetic anisotropy in the plane of the films, reflecting their microstructural texture.
- The designation Ha was introduced earlier (at the beginning of the article) and was used in expression (2) to indicate the anisotropy field. It must be assumed that the same designation here refers to the applied magnetic field. In order not to mislead the reader, it is necessary to clearly formulate the introduced designations.
- The sentence 'It indicates that the interconnected continuous films formed by connect together grains and to generate adjacent magnetic domains in the pure FePt films as shown in Figure 7a' requires clarification and should be stated differently for better understanding.
Author Response
The manuscript can be accepted for publication after some revision containing answers to a number of questions and comments.
- Abstract, introduction and following text mention ‘a high exchange-coupled interaction between the neighboring grains’ ‘strong exchange coupling (positive δM) between neighboring grains in the FePt continuous films’, ‘intergrain exchange coupling between the adjacent FePt nanograins’, and ‘decoupling of the intergranular interaction'. At the same time, the ‘existence of dipole interaction in our designed FePt-MgO composite structure' is discussed. It is necessary to explain more clearly the terminology used here, whether two types of interaction having a different nature are meant (dipole-dipole magnetostatic interaction vs more long-range exchange interaction between individual grains which can hardly be considered reliably established).
Our response:
Thanks for the Reviewer’s kind suggestion. All described sentences with discussion based on the following texts mention (terminology: intergrain exchange coupling) have also been used and modified in red characters in the revised manuscript.
- In the introduction it was stated “Particularly the relationship between micro-/nanostructure and the magnetization process (coercivity mechanism) is discussed in detail for L10ordered FePt-MgO films and related rare earth–transition alloy composited films”, but nowhere else is it said about composite films containing rare earth elements. Apparently, alloys based on transition metals are assumed here, but not RE elements, but Pt.
Our response:
Thanks for the Reviewer’s kind reminder. The texts “rare earth–transition alloy composited films” have been modified as “related anisotropic alloy nanocomposite rare earth permanent films”.
- Has the hysteresis loops in Fig.3 been corrected to account for the demagnetizing field of the continuous film when the films are magnetized in the perpendicular direction?
Our response:
Thanks for the Reviewer’s kind question. Actually, the demagnetization field is insignificant and negligible in a multi-domain sample at our case, and the magnetic crystalline anisotropy plays an essential role in determining magnetization reversal for perpendicular configurations. The similar concept can be found in below references:
- Knudsen and S. Mørup, THE INFLUENCE OF THE DEMAGNETIZING FIELD ON THE MAGNETIC SPLITING OF MÖSSBAUER SPECTRA, Journal de Physique Colloques, 41 (C1), pp.C1-155-C1-156 (1980).
- P. Moore, J. Ferré, A. Mougin, M. Moreno, and L. Däweritz, J. Appl. Phys. 94, 4530 (2003).
- Liu, W. L. Lim, L. V. Titova, M. Dobrowolska, and J. K. Furdyna, J. Appl. Phys.98, 063904 (2005).
- Is there a difference in the magnetization curves depending on the direction of the magnetic field in the plane of the film? This would help to identify a possible magnetic anisotropy in the plane of the films, reflecting their microstructural texture.
Our response:
Thanks for the Reviewer’s kind question. Perpendicular Magnetic Anisotropy (PMA) describes the magnetic anisotropy, in which the direction of easy axes is perpendicular (out-of-plane: marked blue) to the film surface and the direction of the hard axis is parallel (in-plane: marked red) for the film as shown Figure 3. The magnetic misalignment induced by the low-temperature process becomes an unavoidable side effect which is more significant.
Although the (001) texture of FePt structures in XRD patterns (reflecting their microstructural texture) slightly deteriorates with MgO additive as shown Figure 8. The magnetic easy axis is constantly perpendicular to the film plane, and the perpendicular anisotropy is manifest for FePt and FePt-MgO films.
- The designation Hawas introduced earlier (at the beginning of the article) and was used in expression (2) to indicate the anisotropy field. It must be assumed that the same designation here refers to the applied magnetic field. In order not to mislead the reader, it is necessary to clearly formulate the introduced designations.
Our response:
Thanks for the Reviewer’s kind reminder. The anisotropy field (Hi) has been modified and used in expression (2), to differentiate from the applied magnetic field (Ha) for the readers.
- The sentence 'It indicates that the interconnected continuous films formed by connect together grains and to generate adjacent magnetic domains in the pure FePt films as shown in Figure 7a' requires clarification and should be stated differently for better understanding.
Our response:
Thanks for the Reviewer’s kind suggestion. The real meaning is that the surface morphology of the pure FePt film shows the continuous film morphology, which is consistent with SEM and TEM images as shown in Figures 1 and 2. The modified parts have been corrected in red characters in the revised manuscript.

Reviewer 2 Report
The authors of this manuscript studied the effects of the addition of MgO nanolayer on the microstructure Fe/Pt stacked films. They performed magnetic characterizations of Fe/Pt stacked films. This article looks very good, but I have two questions that the authors didn't clearly mention inter the manuscript:
- What is the novelty of this work? How do they work different from other works in this field?
- What is the theoretical model that is supposed to explain the authors' results?
Author Response
The authors of this manuscript studied the effects of the addition of MgO nanolayer on the microstructure Fe/Pt stacked films. They performed magnetic characterizations of Fe/Pt stacked films. This article looks very good, but I have two questions that the authors didn't clearly mention inter the manuscript:
- What is the novelty of this work? How do they work different from other works in this field?
Our response:
Comparing with our previous works [56-57], without any buffer layer and thinner thickness of Fe and Pt stacked layer were designed and assisted in this present study to lower formation temperature of FePt ordered L10 composite transformation at a such low deposition temperature of 380 oC. We demonstrated a simple approach for fabricating nanogranular FePt (001) films with controllable size and coercivity in general. For example, the substrates were heated up to 780 °C for FePt layers and cooled down to room temperature (R.T.) for MgO layers during deposition with very large heating and cooling regions [55]. Our present way (MgO addition nanolayer) is different from the method of co-sputtering technique and better to keep c-axis highly oriented perpendicular to the film plane. The extrinsic metal oxides and nitride elements added into FePt structures were in order to control the c-axis alignment and microstructure of the FePt alloy, which normally caused either random crystal orientation or high transformation temperature for L10 chemical ordering.
- What is the theoretical model that is supposed to explain the authors' results?
Our response:
Shown in Figure 4 are the perfectly theoretical curves, defining two boundary conditions of domain-wall motion and rotation of the Stoner–Wohlfarth (S–W) models, respectively. For a perfect domain-wall motion model, the coercivity at the angle θ is proportional to 1/cos(θ), where θ is the angle between the applied field and easy axis of the uniaxial magnetic anisotropy. As for the S–W model with rotation mechanism, the variation of the coercivity decreases with increasing θ. The angular dependence of the coercivity profile for the FePt without MgO nanolayer showed a typical peak behavior due to the continuous film morphology of the pure FePt films. When the FePt films were added with MgO nanolayers, the profile was close to the rotation mode, and the magnetization reversal behavior become more independent. The reversal process dominated by the nucleation of the reversal domain is rather than the domain wall motion due to the angular coercivity profile of FePt-MgO is closer to the S-W rotation mode.

Round 2
Reviewer 2 Report
After the necessary corrections have been performed, the article is ready for publication.